# Modeling and Optimizing Laser-Induced Graphene

**Lars Kotthoff**     **Sourin Dey**     **Vivek Jain**     **Alexander Tyrrell**     **Hud Wahab**

**Patrick Johnson**
Center for Artificially Intelligent Manufacturing
University of Wyoming
Laramie, WY 82071
{larsko,sdey2,vjain2,atyrrel1,hwahab,pjohns27}@uwyo.edu

## Abstract

A lot of technological advances depend on next-generation materials, such as graphene, which enables a raft of new applications, for example better electronics. Manufacturing such materials is often difficult; in particular, producing graphene at scale is an open problem. We provide a series of datasets that describe the optimization of the production of laser-induced graphene, an established manufacturing method that has shown great promise. We pose three challenges based on the datasets we provide – modeling the behavior of laser-induced graphene production with respect to parameters of the production process, transferring models and knowledge between different precursor materials, and optimizing the outcome of the transformation over the space of possible production parameters. We present illustrative results, along with the code used to generate them, as a starting point for interested users. The data we provide represents an important real-world application of machine learning; to the best of our knowledge, no similar datasets are available.

## 1   Introduction

Graphene is a two-dimensional honeycomb layer of carbon atoms with extraordinary properties, for example relative strength higher than any other material, high conductivity of electricity and heat, and near transparency. It has many promising applications, such as next-generation semiconductors, flexible electronics, and smart windows, to name but a few examples [Ferrari et al., 2015]. There already exist a number of commercially available products made from or with graphene, and the size of the global market is currently about US-$100 million, with significant growth forecast. However, the reliable and large-scale production of graphene is a difficult problem that researchers have been tackling over the past decades.

One method of producing graphene is to convert natural sources of carbon, e.g. graphite, coal, and biochar, into graphene oxide, which is soluble in water. Such solutions can be used as graphene oxide inks and be printed directly onto substrates as thin films, similar to how ink-jet printers deposit ink on paper. Irradiating this precursor material with a laser heats and anneals the graphene oxide selectively to reduce the oxygen, ultimately converting it into pure graphene. Similar results can be achieved by irradiating commercial polymer films, eliminating the need to manufacture and deposit graphene oxide, which is time-consuming in itself, or indeed any carbon precursor material [Chyan et al., 2018]. The reduction of such precursor materials into graphene allows for the rapid and

Submitted to the 35th Conference on Neural Information Processing Systems (NeurIPS 2021) Track on Datasets and Benchmarks. Do not distribute.

chemical-free manufacturing of advanced devices such as electronic sensors [Luo et al., 2016], fuel cells [Ye et al., 2015], supercapacitors [Lin et al., 2014, El-Kady and Kaner, 2013], and solar cells [Sygletou et al., 2016]. The interested reader is referred to a recent survey on laser-induced graphene for more information [Wang et al., 2018]. This process is also referred to as laser-reduced graphene in the literature [Wan et al., 2018].

One of the advantages of the targeted irradiation of the precursor material is that it allows to easily create patterns in solid substrates without pre-patterned masks in only a few minutes. While graphene is electrically conductive, graphene oxide and polymers are not – patterns of graphene in an insulating material can form electric circuits. The laser irradiation process enables the scalable and cost-efficient fabrication of miniaturized electronic devices in a single process, rather than manufacturing the graphene separately and then patterning it onto a carrier material. This process also ensures that only the amount of material that is actually needed is produced, similar to other advanced manufacturing processes like 3D printing.

The challenge in irradiating the precursor material is determining the best laser parameters and reaction environment. First-principles knowledge does not allow to derive the optimal conditions and the effectiveness of different irradiation conditions varies across different precursor materials. A recent study emphasizes the effect the irradiation parameters have on the quality of the produced graphene and the need to optimize these parameters to achieve good results in practice [Wan et al., 2019]. Even with just a few parameters, for example the power applied to the laser and the duration for irradiating a particular spot, the space of possibilities is too large to explore exhaustively. There are complex interactions between parameters, and evaluating a particular parameter configuration involves running an experiment that requires a skilled operator and precursor material of sufficient quality. Exploring the space of experimental parameters efficiently is crucial to the success of laser-induced graphene in practice. In many cases, this optimization is guided by human biases – an area ripe for the application of machine learning.

We have applied Bayesian optimization to the automated production of laser-induced graphene, improving the quality of the produced graphene significantly compared to results achieved in the literature [Wahab et al., 2020]. In this paper, we present a series of datasets obtained in the process for the community to build on. To the best of our knowledge, there are no similar datasets. In particular, the data we make freely available represents an important and challenging application of machine learning in a rapidly-growing industry. Beyond graphene, materials science in general is an increasingly prominent application area of machine learning. We outline possible uses for the data, along with illustrative results. All data, code, and results are available at `https://github.com/aim-uwyo/lig-model-opt`.

## 2   Methodology

The graphene oxide samples used for the data we present here were prepared from graphite using the improved Hummers' method [Marcano et al., 2010]. Powdered samples, ground and sieved to $20\,\mu m$, were mixed in concentrated $H_2SO_4$ and $H_3PO_4$ and placed in an ice bath. $KMnO_4$ was added at a mixture temperature of $35\,°C$ and increased to $98\,°C$ before termination with ultrapure water (Millipore) and $H_2O_2$. The filtrate was then washed with $HCl$ and subsequently with water repeatedly until a pH-level of about 6.5 was obtained. The GO inks were produced using $25\,mg$ of the freeze-dried GO powder, which was diluted in $100\,ml$ deionized water and ultrasonicated with a cooling system. After the sample was centrifuged, the remaining supernatant was repeatedly diluted and ultrasonicated until a $200\,ml$ dilution was obtained. The GO inks were spray-coated onto a $1\,cm{\times}1\,cm$ quartz or polyimide substrate (Kapton HN $125\,\mu m$, Dupont) in multiple passes until a thickness of $1\,\mu m$ was achieved, verified with an optical profilometer.

Laser-induced graphene (LIG) spots were patterned by reducing GO films deposited on quartz and polyimide, and by carbonization of polyimides directly. We denote GO on quartz, GO on polyimide and polyimide as samples GOQ, GOPI and PI, respectively. The patterning setup is shown in Figure 1. The deposited GO films were placed in a sample chamber which allows patterning in air, argon, or

nitrogen environments with pressures up to 1000 psi. LIG patterns were irradiated using a 532 nm diode-pumped solid-state continuous-wave laser. The laser beam was focused with a 50x microscope lens to a spot size of 20 μm on the sample surface. Irradiated beam spots were positioned sufficiently far apart from each other to ensure pristine precursor material for each experiment. The sample area is about $1\,\mathrm{cm}^2$, allowing approximately 256, 25, and 25 patterns for samples GOQ, GOPI, and PI, respectively. Taking into account sample preparation and repeated measurements to account for experimental errors and ensure reproducibility, we set our experimental budget to 70 for all types of samples.

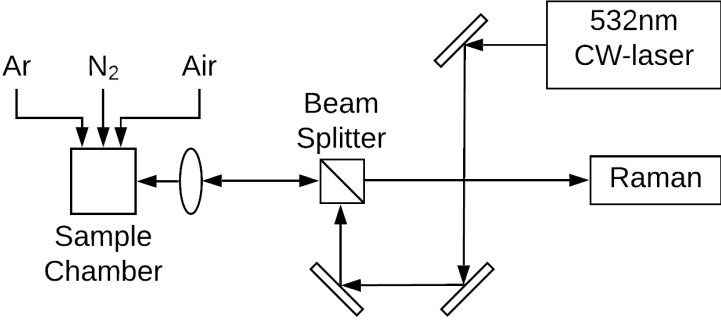

Figure 1: Experimental setup for patterning and measuring laser-induced graphene. The unlabeled rectangles represent mirrors to reflect the laser beam, the ellipse a lens to focus it.

Raman spectroscopy is a common technique for determining the quality of laser-induced graphene by observing how laser photons scatter after they interact with the vibrating molecules in the sample probe. The intensities of the characteristic D and G bands in the Raman spectra can be used to judge to what extent the precursor material has been reduced to graphene, i.e. the quality of the resulting material. The D and G bands result from the defects and in-plane vibrations of $\mathrm{sp}^2$ carbon atoms, respectively. In particular, the degree of reduction of the precursor material to graphene, and thus the conductivity of the irradiated area, can be quantified through the ratio of the intensities of the G and D bands – the larger this ratio, the more the precursor material has been reduced. Figure 2 shows an example.

We filtered the backscattered laser beam through a long-pass filter after irradiation to perform Raman spectroscopy. Using the same laser source for patterning and spectroscopy, we are able to characterize the identical spot in-situ. The Raman data for each spot were averaged over 10 measurements with a collection time of 3 s at laser power <10 mW for each measurement. The Raman spectra were post-processed with a linear background subtraction to 0 and normalization of the maximum peak to 1. The G- and D-bands were fitted using Lorentzian functions and the ratio of their intensities computed as the ratio of the areas under the fitted functions. The G/D ratios indicate the degree of reduction of GO to graphene. This measure can be used as a proxy for electric conductivity, which determines the suitability of the produced material for advanced electronics. More information on the experimental setup can be found in [Wahab et al., 2020].

## 2.1 Parameter Space

We consider the following four parameters of the experimental conditions that control the irradiation process.

- The power applied to the laser used to irradiate the sample. We consider a power range of 10 mW to 5550 mW.

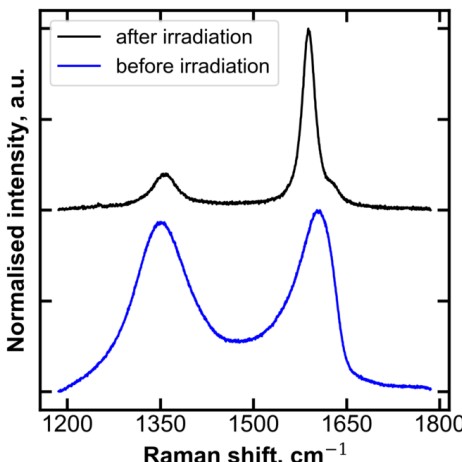

Figure 2: Raman spectra showing D (left peak) and G (right peak) bands of graphene oxide before (bottom) and after (top) laser irradiation. The ratio we optimize in this paper is calculated from the area under the peaks. The intensity is shown in arbitrary units (a.u.).

- The duration a particular spot was irradiated by the laser. We vary this parameter from 500 ms to 20 000 ms.

- The pressure in the reaction chamber. The values for this parameter range from 0 psi to 1000 psi.

- The gas in the reaction chamber. Possible values for this parameter are argon, nitrogen, and air.

These parameters give rise to a large space of possible combinations that is infeasible to explore exhaustively. The cost of gathering data is high – running experiments is time-consuming and requires precursor materials to be available. In contrast to big-data approaches, we need techniques that work with small amounts of data, such as the Bayesian optimization approach we applied to gather the data we present here.

## 2.2 Bayesian Optimization

Bayesian model-based optimization techniques (MBO) are used in many areas of machine learning and AI and beyond to automatically optimize outcomes across large parameter spaces. They usually proceed in an iterative fashion – they predict the configuration to evaluate, and the result of this evaluation informs the predictions for the configuration to evaluate next. At the heart of these techniques are so-called surrogate models, which approximate and model the process whose parameters are to be tuned. This underlying process is expensive to evaluate, i.e. it is infeasible to exhaustively explore the parameter space and we are interested in keeping the number of evaluations as small as possible. The approximate surrogate model on the other hand is cheap to evaluate and allows for a targeted exploration of the parameter space, identifying promising configurations that available resources for evaluations of the underlying process should be directed towards.

Surrogate models are induced using machine learning, taking an increasing amount of ground-truth data into account between subsequent iterations. State-of-the-art MBO approaches often use Gaussian Processes or random forests to induce surrogate models, depending on the nature of the parameter space. MBO is a mature approach that has been used in many applications over decades, for example in automated machine learning [Feurer et al., 2015, Kotthoff et al., 2017]. The interested reader is referred to the paper that formalized the approach [Jones et al., 1998] for more information.

There are many implementations of MBO; we use the mlrMBO package [Bischl et al., 2017] to model the parameter space, build the surrogate models (with the mlr package [Bischl et al., 2016]), and determine the most promising configuration for the next evaluation of the underlying process. In particular, we use the default random forest surrogate model for parameter spaces that contain non-continuous parameters (the gas in the reaction chamber) and expected improvement as our acquisition function. In each iteration of the optimization process, the next configuration to evaluate is proposed by mlrMBO. This configuration is set automatically by the experimental setup, which proceeds with running the experiment and evaluating its result. The evaluated parameter configuration and the resulting G to D ratio of the irradiated spot is added to the data used to train the surrogate model for the next iteration. We present the datasets obtained when the process ends.

For the initial surrogate model, we evaluated 20 parameter configurations that were randomly sampled from the entire parameter space. We then performed 50 iterations of our model-based optimization approach, for the total 70 evaluations we can perform on a single sample. For each of the three investigated materials GOQ, GOPI, and PI, we ran three experimental campaigns for a total of nine experimental campaigns and 630 patterned spots, which represents several weeks of experimental effort, in addition to the effort of preparing the samples.

## 3 Machine Learning Datasets from Materials Science

We present three datasets from the above application. All datasets describe the transformation from a precursor material into graphene through laser irradiation, where the quality of the result depends on the parameters of the laser and reaction environment. The difference between the different datasets is that they were obtained with different precursor materials, as described above. Each dataset consists of three experimental campaigns each, for a total of 210 data points per dataset. Metadata is provided for each datum indicating which experimental campaign it belongs to and whether it was part of the initial, randomly sampled data, or proposed by the Bayesian optimization process.

We envision three different areas of machine learning where this data will be useful; we describe each below with illustrative results and the code that was used to produce them. Our illustrative results are intended to show what performance can be achieved to give prospective users a starting point; we do not make any claims with respect to the optimality of our results.

The data, scripts to produce the illustrative results we describe below, and the figures themselves are available at `https://github.com/aim-uwyo/lig-model-opt` under the permissive 3-clause BSD license. No ethical issues arose in gathering the data, but we caution that they could potentially be used in unethical applications, for example to produce advanced electronics for weapons systems. We do not condone or encourage such applications.

### 3.1 Modeling Laser-Induced Graphene

The datasets we provide can be used in straightforward manner to predict the quality of the transformation of the precursor material into graphene, given the experimental parameters. We ran illustrative experiments with the mlr3 machine learning toolkit [Lang et al., 2019]. The code necessary to reproduce our results is provided in supplementary material, together with the data. In addition, we ran experiments with the auto-sklearn [Feurer et al., 2015] automated machine learning toolkit, running it with a time limit of one hour.

We present illustrative results in Figure 3. Even simple approaches, such as linear models, already achieve much better performance than the baseline featureless learner. More sophisticated approaches, such as the Gaussian Processes and random forests that are ubiquitous surrogate models in Bayesian optimization, do not further performance much. The same is true for much more sophisticated automated machine learning approaches, especially on the GOQ dataset.

Results are best for the GOPI and PI datasets in terms of improvement over the baseline, with GOQ showing a smaller gap. We believe that model performance can be increased further; in particular, automated machine learning has been able to improve performance only slightly here.

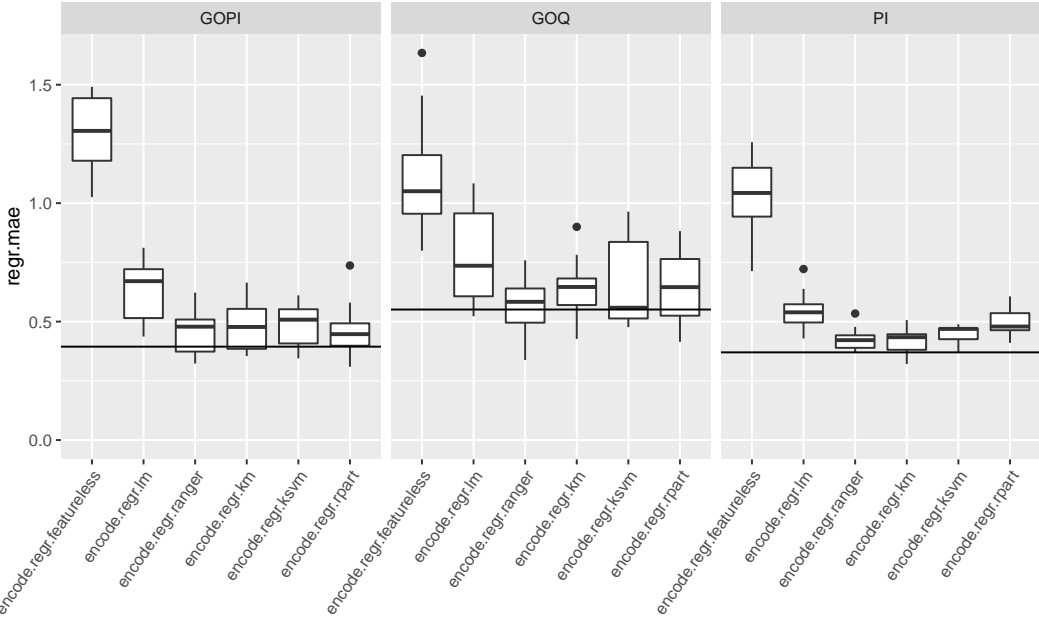

Figure 3: Illustrative results for modeling the transformation of the precursor material into graphene for different machine learning approaches. We show the mean absolute error over 10 cross-validation folds for (from left to right) a dummy featureless learner, a simple linear model (lm), a random forest (ranger), a Gaussian Process (km), a support vector machine (ksvm), and a regression tree (rpart). The featureless learner simply predicts the mean value of the training set. The horizontal lines denote the performance of auto-sklearn on each of the datasets.

We note that, in contrast to most machine learning datasets, the data we present here is not identically and independently distributed, as it has been obtained as part of Bayesian optimization runs, where a data point depends on the previous ones. This violates the basic assumption underlying most machine learning approaches. In practice, building surrogate models from non-i.i.d. data appears to work fine, as good results from applying Bayesian optimization, including ours, show. Nevertheless, the implications of using non-i.i.d. data in this context are understudied, and our data provides and opportunity to do so. Each point in the raw data has metadata denoting whether the point was obtained as part of the initial, random and i.i.d., data or evaluated in a Bayesian optimization iteration.

## 3.2 Transfer Learning

Three datasets from very similar but different setups also provide the opportunity to explore to what extent knowledge acquired from one dataset can be transferred to another. While the process is the same, the precursor materials are different and react differently to the same experimental conditions. There are latent features that encode the properties specific to each precursor material that machine learning may be able to extract.

We ran illustrative experiments, again with the mlr3 machine learning toolkit. Figure 4 shows illustrative results from the evaluation of a model learned on one dataset on another. It is immediately clear that the two precursor materials based on polyimide, GOPI and PI, behave very similarly – models trained on one precursor material transfer with good performance to the other, although not as good as for models trained and evaluated on the same dataset (two middle panels in the figure). For GOQ, transferred models (both from and to this precursor material) do not show good performance compared to the baseline model, indicating that the precursor materials are sufficiently different that a direct transfer is infeasible.

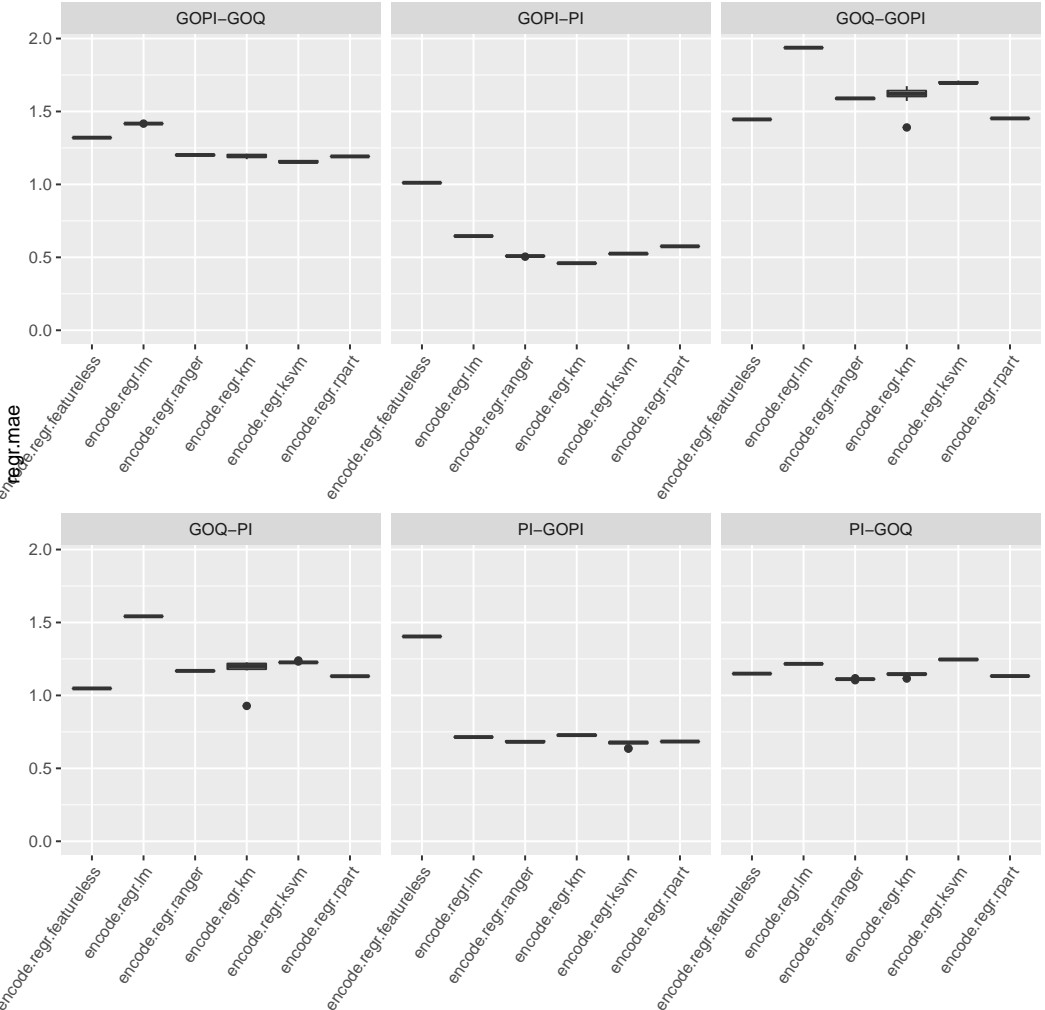

Figure 4: Illustrative results for transferring models from one precursor material to another. We show the mean absolute error for the same learners as above, including the dummy featureless learner. The first dataset in the title of a plot denotes the training set, while the latter denotes the test set. We randomly sample 80% of the respective datasets for training and test, repeated 10 times.

We further explore the performance of transferred models in Figure 5, this time by training models on the combination of datasets of two precursor materials and evaluating their performance on the dataset of the third precursor material. We again see that the two precursor based on polyimide are quite similar, while GOQ is different and transferred models do not exhibit good performance.

We provide two datasets that are quite similar, GOPI and PI, and one that is quite different from the others, GOQ. This allows to create "easy" and "hard" transfer learning scenarios.

## 3.3 Bayesian Optimization

The datasets we provide can also be used for Bayesian optimization, which is how the data was obtained to start with. In the end, we are interested in the best conversion of precursor material into graphene and better ways of obtaining the experimental parameters for that. An interesting aspect of the data we provide is that it comes from a real-world application with a good motivation for applying a sample-efficient optimization method, as obtaining new data points is extremely expensive.

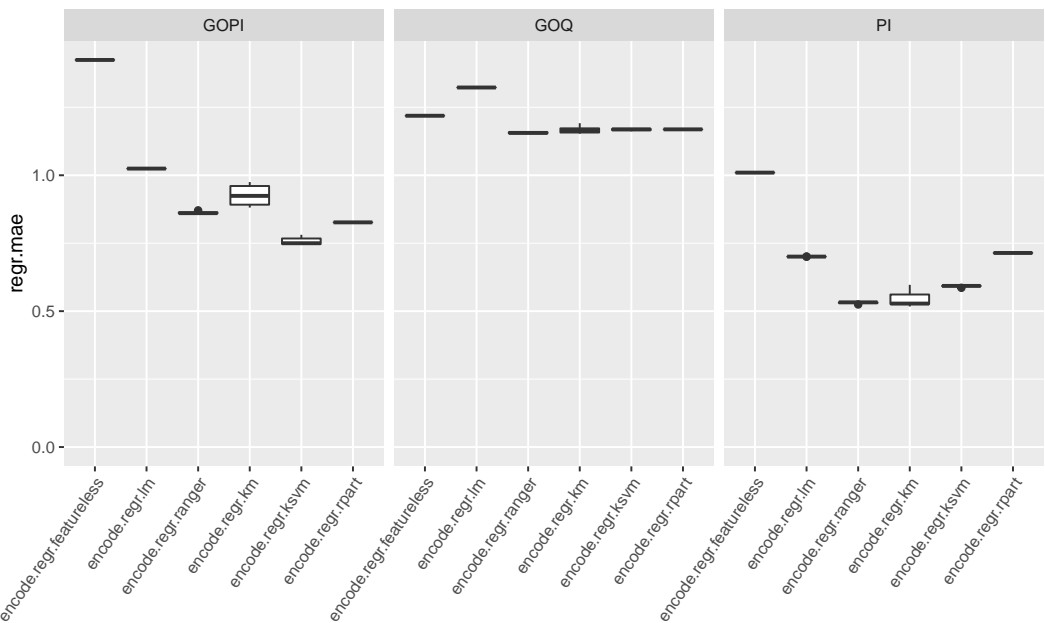

Figure 5: Illustrative results for transferring models trained on two precursor materials to the other. We show the mean absolute error for the same learners as above, including the dummy featureless learner. The title of the dataset denotes the one that the performance of the models learned on the other two was tested on. We randomly sample 80% of the respective datasets for training and test, repeated 10 times.

We provide simulators based on surrogate models built on entire datasets to facilitate Bayesian optimization. Simulators and illustrative experiments are based on the mlr [Bischl et al., 2016] and mlrMBO [Bischl et al., 2017] toolkits; the same we used in the publication related to our datasets.[1] Figure 6 shows illustrative results. We show only results for the GOPI precursor material for space reasons; results for the other precursor materials are qualitatively similar and available at `https://github.com/aim-uwyo/lig-model-opt`. Interested users can easily plug in their own approach and evaluate how efficiently and effectively it explores the optimization landscape provided by the surrogate models.

There are multiple ways our data can be used to improve the Bayesian optimization process. Better surrogate models will enable better optimization, and can be explored independently. Similarly, being able to transfer knowledge from other Bayesian optimization runs, for example on different precursor materials, will improve performance. Both of these challenges can be pursued with the datasets we provide, in addition to methodological improvements to Bayesian optimization.

Explaining black-box machine learning models is becoming increasingly important, especially for real-world applications like the one we present here. On one hand, being able to understand a model increases trust in it, while on the other hand a machine-learned model may have acquired insights that are unknown to humans and may advance our scientific understanding of the optimized process. We explore some such methods in [Wahab et al., 2020], but there is scope for further exploration and explanation of the surrogate models.

We note that the simulators we provide can be used to evaluate different optimization methods, such as genetic algorithms or Hyperband, equally as well. We focus on Bayesian optimization here as this is the methodology we applied for the application itself, but what we provide is not limited to that.

---

[1]While the mlr toolkit has been superseded by mlr3, the corresponding successor to mlrMBO is not available yet. The underlying machine learning algorithms are the same.

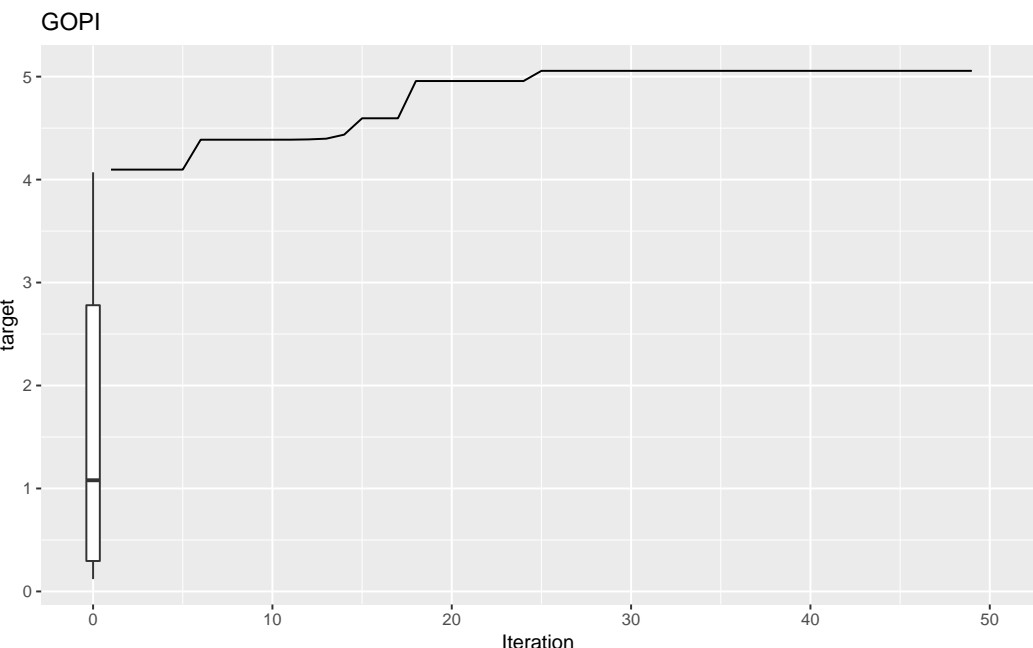

Figure 6: Illustrative results for Bayesian optimizations on simulators trained on entire datasets, here for the GOPI precursor material. The boxplot at iteration zero shows the distribution of the initial, randomly sampled data, while the line shows the cumulative best achieved transformation from the precursor material into graphene (measured by the G/D ratio) over the iteration number of the Bayesian optimization.

## 4 Conclusions and Outlook

We have presented three datasets drawn from a real-world application of machine learning; the production of laser-induced graphene. The data are accompanied by metadata and example code that demonstrates possible uses. To the best of our knowledge, it is the first series of datasets from materials science with the presented level of comprehensiveness, and we hope that it will facilitate and inspire more applications of machine learning in this area and beyond.

Gathering the data we make available took significant effort, from preparing the samples, running the experiments, to post-processing the raw experimental data. This is common in materials science, where gathering data often involves synthesizing a material or performing an experiment that leads to its transformation or destruction. For this reason, big data methods are not applicable here, or may only be applied with difficulty. We hope that by making our data available, we will stimulate research on small data and sample-efficient methods.

The code used to obtain the illustrative results we present here is available as part of the datasets, and all results are fully reproducible. This provides an easy starting point for interested users. We place no restrictions on the use of the code and data we make available, but discourage unethical uses.

**Acknowledgments**

We are supported by the University of Wyoming's College of Engineering and Applied Sciences' Engineering Initiative, the School of Energy Resources at the University of Wyoming, the Wyoming NASA Space Grant Consortium, and NASA EPSCoR. SD and LK are supported by NSF award #1813537. The sponsors had no involvement in the creation of the datasets or this manuscript.

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

# A Appendix

The code and data we reference in this paper are available at `https://github.com/aim-uwyo/lig-model-opt` under the 3-clause BSD license. Raw data is provided in CSV files, which can be widely read without problems. Code and example outputs are provided in plain text files and PDF files for figures. The README file of the repository explains each file and gives more details on the format of the CSVs. The provided code fixes random seeds to ensure reproducibility of results. Data sources and methods used for obtaining the data are described in this paper.

We have submitted a request for a dataset nutrition label and will add it as soon as the request is processed. After acceptance of the paper we will create a release in the above repository and get a DOI for it; we have not done this so far to allow for changes that the reviewers may suggest to be incorporated. The intended uses of our data are outlined in the paper; we envision it being used for modeling, transfer learning, and optimization.

The authors bear all responsibility in case of violation of rights etc. The license the datasets are provided under is the 3-clause BSD license. The repository will be maintained by the authors, with GitHub's issue and pull request system allowing users to ask questions, raise issues, and suggest changes.

