# OpenReview forum: "Modeling and Optimizing Laser-Induced Graphene"
_NeurIPS.cc/2021/Track/Datasets_and_Benchmarks/Round1 — Submitted to NeurIPS 2021 Datasets and Benchmarks Track (Round 1)_

### Official Review · Reviewer_HAET · 2021-07-03
**Review for Modeling and Optimizing Laser-Induced Graphene**

**Rating:** 4
**Confidence:** 4
**Clarity:** This paper is well written, github we…

**Strengths:**

The major contribution is in the dataset it publishes. It has 630 data points, with 4 controlled parameter( power	time	gas	pressure ) and 1 measured property (target). The process of getting the experiment done seems well documented  and the overall paper is clear and easy to follow.

The 3 ML related experiments using existing packages is nice and clean. The 10 fold cross val is reasonable and robust given the small sample size of each dataset (210 points). The objective is well understood for each experiment and graphs are easy to understand as well.

**Weaknesses:**

First of all, as this is NeurIPS, this review is constructed more towards machine learning than material science.

Significance of contribution:

The significance of this new dataset is pretty marginal to me. Although this is a well-constructed datasets with rigourous experiments being done in the lab, I am not convinced that this dataset is largely different from online datasets people can get on Kaggle or UCI database.

I agree that for material scientists LIG might be of great interest and hence very important to have public datasets available to develop new techniques, but for the ML community, this specific application is not likely to be transferred into other areas given the size of this dataset.

More specifically, quantity and quality are two factors that I cherish for datasets. Quantity: 210 datapoints for one precursor is not enough for people to build big models to completely model it (Just as the Fig 3, the best model found by autoML still has a ~0.4 MAE, which is just 1/3 of a random guess). And if people can not have "simulator" (very accurate models), even if they can try their MO methods, there is no way to actually know how they are performing, assuming they do not have the resource to actually do laser experiments. Quality: This specific application might be novel and the first publicly available dataset, but how is this different from publishing their dataset along side their journal publication, lets say a paper (Design of Experiments and Optimization of Laser-Induced Graphene by Murray et.al.) publishes its log book?


Revevance to the broader research community:
To my mind, the (offline) Model-based optimization community is the major target audience of this dataset. However, without a accurate enough simulator that actually captures the real forward function, it is hard for them to evaluate their MO methods. For this, maybe you can do the same trick as Ren et.al (Benchmarking deep inverse models over time, and the neural-adjoint method, NeurIPS2020) by constructing a accurate enough model, however I do think the data bottleneck is an issue there.


IID problem:
As the authors themselves have pointed out, iid plays an important role in the ML community. This dataset, largely consists of non-iid sampled datapoints from the baysian optimization is much less to me as a carefully constructed public dataset for other researchers to build on but submitting the experiment log of a previous paper (Machine-learning-assisted fabrication: Bayesian optimization of laser-induced graphene patterning using in-situ Raman analysis).

Transfer learning problem:
Maybe I did not understand this part fully...To me, transfer learning is the process of taking weights and biases (or parameters) learned from other domains as a starting point (instead of random initialization) of your new domain learning tasks and then learn (finetune) your model upon the actual learning task you are working on.
Here, how the authors presents this part is more like a "Cross-domain testing" instead of transfer learning as there are no learning after the knowledge being transferred from the other domain.

Line 166: "(The third area) of machine learning where this data would be useful" -- Bayesian optimization
I agree that doing a BO on this dataset is useful for designing new LIG processes. However , I failed to understand how this data can help other people do any BO experiment given the bad performance of the "simulator".

Suggestions: When the distribution of data is not iid, especially in an offline model based optimization realm, the distribution of target variable, the distribution of the prediction error w.r.t. target variable is highly important in the training set. More details like this would be strengthning the paper.

**Additional Feedback:**

Minor details (questions):
Line 208 (two middle panels in the figure), where exactly is the in-domain testing result apart from Fig 3?
Line 195: typo: our data provides (an)d opportunity...

Overall, the paper is nice and clean, however from my perspective this is not enough for NeurIPS.

Updated comments upon reviewing rebuttal and other reviewers' comment: (Jul 19th)
Thank the authors for the extra information provided. I agree that smaller dataset also have high values for the general ML community to work on (they are challenging and important problems to solve), however the non iid distribution makes it nearly impossible for methods to be applied and compared fairly. Therefore I am remaining my origin evaluations.

**Correctness:**

The claims made in the paper are correct to me. The dataset is not constructed iid and the amount of data is not likely to be widely used in the ML community for algorithmic development but (possibly) enough for the LIG community.

The evaluation methods and experimental design are correct barring the suggestions made in the "Weaknesses" part.

**Documentation:**

Yes, sufficient details on collection, organization, availability and maintenance, ethical and responsible use is included.

**Ethics:**

No ethics concerns found.

**Relation To Prior Work:**

As a dataset proposing paper proposing a novel dataset, I do not think this discussion is particularly important.

**Summary And Contributions:**

This paper proposed (publishes) a new dataset which did experiments for Laser-induced graphene (LIG) on 3 precursors (GOP, GOPI and PI). It has 630 experiments (thus, 630 datapoints), each precursor has 210 with 90/210 randomly iid sampled and the rest being part of a bayesian optimization run.

This work also provided 3 potential "different areas of machine learning where this data would be useful":
1. Forward modelling (From controlled parameters to property, or target value)
2. "Transfer learning" (Forward modelling but using slightly different domain of data)
3. "Bayesian Optimization" (Inverse design, using BO to find the optimal design, as defined by high D/G ratio in Ramen spectroscopy.

The main contribution, or novelty of this paper is the new dataset of 630 points for LIG process.

---

### Official Review · Reviewer_e6K6 · 2021-07-04
**Method for optimizing laser-induced graphene**

**Rating:** 5
**Confidence:** 2

**Strengths:**

- The authors proposed to use Bayesian optimization to more effectively explore the parameter space when productionizing laser-induced graphene. The problem is well-motivated and the adopted solution is also reasonable.

- The proposed datasets are shown to have multiple use cases: 1) the datasets can be used to predict the transformation quality from precursor material into graphene; 2) the datasets can be used in transfer learning to obtain knowledge which precursor material can better transfer to the other; 3) for further optimization to improve sample efficiency.

**Weaknesses:**

- One main concern I have is the novelty of the current work. The paper seems to have applied Bayesian optimization in determining the best parameters in the production process of laser-induced graphene, which is similar to the method proposed in [Wahab et al., 2020]. A detailed discussion on the how the current work differs from existing work is needed.

- I don't quite understand the paragraph on model explanation (line 237-242), is there any experiment performed to explain the black-box models in the current application? This paragraph seems to be a discussion on the possibility rather than providing any insights.

- Most of the application cases mentioned in this paper seem to merely present the empirical results but without any practical suggestions. For example, in section 3.1, given most modeling approaches perform similarly on all datasets (except the dummy featureless learner), what does this suggest for practitioners on which algorithm to use? Also, is there a reason more sophisticated methods do not further the performance much?

**Additional Feedback:**

Please include a more detailed explanation on how the proposed use cases can be helpful for practitioners in making better decisions when working in material science.

**Clarity:**

The paper is mostly clear.

Figure 5, caption, "We randomly sample 80% of the respective datasets for training and test", do you mean you sampled 80% for training and 20% for test?

**Correctness:**

The ML part of the paper seems mostly correct to me, but I'm not very familiar with the terminology in material science so there could be things that I missed.



**Documentation:**

Most of the experimental details are described well in the paper.

The authors did not mention ethical or responsible use of the proposed datasets. No maintenance plan is mentioned.

**Ethics:**

No.

**Relation To Prior Work:**

Some connection to prior work was discussed but it is unclear to me what is the main difference/novelty of the current work.

**Summary And Contributions:**

This paper applied Bayesian optimization for automating the production of laser-induced graphene, as the main challenge in creating such datasets is finding the best laser parameters and reaction environment. A series of datasets are presented to facilitate the research in material science. A few application cases based on the proposed dataset are also proposed, including predicting the quality of modeling approaches, for transfer learning to obtain knowledge between different precursor materials, as well as further Bayesian optimization to improve sample efficiency.

---

### Official Review · Reviewer_5zbe · 2021-07-06
**This paper uses Bayesian optimization to explore optimal experimental configurations for producing laser-induced graphene. They release the associated dataset and also train 'simulators' that can output the quality of produced sample given input of experimental configuration.**

**Rating:** 4
**Confidence:** 4

**Strengths:**

The details for real-world generation of graphene and the computational procedure of Bayesian optimization are provided carefully. The dataset with relevant code and graphs is made available. The collection of this data involves producing graphene with 600+ distinct design knobs, which is a meticulous procedure.

The paper also discusses ways of utilizing this dataset for improving/exploring better surrogate models and transferring the knowledge from model(s) trained on one dataset to another dataset doing a similar experiment with different starting materials.

**Weaknesses:**

The paper does not introduce a novel idea within the domain of machine learning. I see that Lars Kotthoff et al., “AI for Materials Science: Tuning Laser-Induced Graphene Production" discuss the very same approach (minus the sections on transfer learning and simulators). I understand that this paper releases a dataset corresponding to these experiments and suggests ways in which this could be useful for the machine learning community. But the sections on 'Transfer Learning' and 'Modeling Laser induced graphene' do not discuss the unique ways in which this dataset helps (I can see that there are graphs showing the performance of different models and how model trained on one dataset can be used to simulate for another one but I believe that we need more details on how this dataset uniquely helps ML/Materials Science researchers).



**Additional Feedback:**

The use of this dataset to create simulators and transfer the learnings to different models can be a good application, but it could be helpful to discuss more precisely how this dataset can aid evaluation of optimization strategies, or how does it help in selection of better surrogate models (from the perspective of machine learning domain).

Edit: Thanks for your response! I appreciate that the dataset has been collected meticulously by running these experiments for every configuration selected by the Bayesian optimization module. This can be a valuable asset for Materials Science research.
At this stage, I feel that we need more experiments/discussion that speaks about how this dataset can uniquely help ML community. At the moment, it seems that the 'simulator' model will not have enough data to actually simulate the output of graphene production in the sections of input domain that are under-explored.

Overall this paper is well-written. The techniques are explained nicely. However, from my point of view, I feel that this may not be enough for NeurIPS.

**Clarity:**

The experimental result graphs can be polished by describing the axes more concisely. The formatting of these graphs can be improved. The experimental section does not have enough details on how the simulator models were selected and trained, and how they might be precisely used to evaluate other optimization strategies. Similarly, the precise ways in which the transfer learning idea can be generalized to different tasks within Materials Science is not sufficiently explained.

**Correctness:**

I believe that the released code and dataset looks correct.
The codebase includes details for all the graphs included in the paper and a clear description of the various components. It is not possible to check for correctness of the actual output values (reflecting quality of graphene produced).

The evaluation in terms of mean absolute error to compare between different 'simulator' models and transfer learning tasks is okay. But the paper does not include details on how the models were selected, tuned and trained.

**Documentation:**

Yes, I feel that they provide sufficient details on the data collection and briefly discuss the ethical considerations.
The dataset is made available with requisite code and csv files, though the file organization and readme file could be more polished.



**Ethics:**

It could be helpful to make sure that these experimental configuration knobs are not used to manufacture harmful chemicals. The paper mentions that they don't condone or encourage such applications.

**Relation To Prior Work:**

This work seems to build on Lars Kotthoff et al., “AI for Materials Science: Tuning Laser-Induced Graphene Production by releasing the dataset and exploring ways in which it can benefit the machine learning community.  But this is not cited in the paper and only appears on the GitHub page.

The paper mentions that there are no similar datasets available within this area but it would be nice to have a separate section the discusses the key contributions and comparison of this work with pre-existing work.

**Summary And Contributions:**

The paper applies Bayesian optimization to explore better experimental knobs for production of laser-induced graphene. The details of the methodology to produce chemical graphene is provided carefully. They give the details of Bayesian optimization setup and the output dataset that is generated. The dataset consists of the parameter settings explored by Bayesian optimization (power, time, gas, pressure) and the corresponding output quality of the graphene samples produced in the real-world experiment. It is clear that this dataset collection involves meticulously running a real-world experiment to produce graphene under the parameter settings picked by Bayesian optimization (along with the initial parameter values that were chosen randomly).

The paper also talks about creating 'simulators', i.e. machine learning models that are trained on this dataset. The corresponding cross-validation errors are noted and the use of these simulators to further evaluate/improve optimization for this application area is suggested.

---

### Decision · Program_Chairs · 2021-07-27

**Decision:**

Reject

**Comment:**

The authors apply Bayesian optimization to the production of graphene and present datasets that facilitate material science research. The reviewers noted the innovative application of Bayesian optimization, but raise concerns about the novelty and the significance of this work and note overlap with previous work. Because of these concerns, the reviewers recommend against accepting the paper